# Emergence of a Novel G4P[6] Porcine Rotavirus with Unique Sequence Duplication in NSP5 Gene in China

**DOI:** 10.3390/ani14121790

**Published:** 2024-06-14

**Authors:** Xia Zhou, Xueyan Hou, Guifa Xiao, Bo Liu, Handuo Jia, Jie Wei, Xiaoyun Mi, Qingyong Guo, Yurong Wei, Shao-Lun Zhai

**Affiliations:** 1Guangdong Provincial Key Laboratory of Livestock Disease Prevention, Institute of Animal Health, Guangdong Academy of Agricultural Sciences, Scientific Observation and Experiment Station of Veterinary Drugs and Diagnostic Techniques of Guangdong Province, Guangzhou 510640, China; xiazhou698868@163.com (X.Z.); 18083906766@163.com (X.H.); xgfmicro@outlook.com (G.X.); 18842550820@163.com (B.L.); jiahanduo@163.com (H.J.); 2College of Veterinary Medicine, Xinjiang Agricultural University, Urumqi 830052, China; wlmqgqy@126.com; 3Xinjiang Key Laboratory of Animal Infectious Diseases, Institute of Veterinary Medicine, Xinjiang Academy of Animal Sciences, Urumqi 830013, China; xiaojie0717@aliyun.com (J.W.); mixiaoyun1220@163.com (X.M.)

**Keywords:** rotavirus, porcine, zoonosis, duplicated sequence, genomic rearrangement, NSP5

## Abstract

**Simple Summary:**

Rotavirus is an increasingly widespread zoonotic agent, the segmented feature of rotavirus genome is responsible for the emergence of new diverse viruses and inter-species transmission. Therefore, monitoring the genetic diversity of rotavirus is helpful to understand its evolution. In this study, a novel potential zoonotic G4P[6] rotavirus with a 344-nt duplication sequence in the non-encoding gene of NSP5 was for the first time detected in a diarrhoea pig sample collected from Guangdong Province, China. Further genome analyses revealed genome reassortment between human-origin and porcine-origin rotavirus.

**Abstract:**

Rotavirus is a major causative agent of diarrhoea in children, infants, and young animals around the world. The associated zoonotic risk necessitates the serious consideration of the complete genetic information of rotavirus. A segmented genome makes rotavirus prone to rearrangement and the formation of a new viral strain. Monitoring the molecular epidemiology of rotavirus is essential for its prevention and control. The quantitative RT-PCR targeting the NSP5 gene was used to detect rotavirus group A (RVA) in pig faecal samples, and two pairs of universal primers and protocols were used for amplifying the G and P genotype. The genotyping and phylogenetic analysis of 11 genes were performed by RT-PCR and a basic bioinformatics method. A unique G4P[6] rotavirus strain, designated S2CF (RVA/Pig-tc/CHN/S2CF/2023/G4P[6]), was identified in one faecal sample from a piglet with severe diarrhoea in Guangdong, China. Whole genome sequencing and analysis suggested that the 11 segments of the S2CF strain showed a unique Wa-like genotype constellation and a typical porcine RVA genomic configuration of G4-P[6]-I1-R1-C1-M1-A8-N1-T1-E1-H1. Notably, 4 of the 11 gene segments (VP4, VP6, VP2, and NSP5) clustered consistently with human-like RVAs, suggesting independent human-to-porcine interspecies transmission. Moreover, a unique 344-nt duplicated sequence was identified for the first time in the untranslated region of NSP5. This study further reveals the genetic diversity and potential inter-species transmission of porcine rotavirus.

## 1. Introduction

Rotavirus (RV), initially identified in the United States in 1969 [1], is a zoonotic pathogen associated with severe diarrhoea and dehydration in children under the age of five years and various young mammals and birds [2]. Especially in low- and middle-income countries, RV constitutes a significant disease burden. In China, the RVA-positive rates were nearly 20% in children and 3.51–7.17% in adults, respectively [3,4].

RV, belonging to the genus of *Rotavirus* within the family *Reoviridae*, has a genome comprising 11 segments of double-stranded RNA that encode 6 structural viral proteins (VP1, VP2, VP3, VP4, VP6, and VP7) and 5/6 nonstructural proteins (NSP1, NSP2, NSP3, NSP4, and NSP5/6). RV consists of a triple-layered capsid, with VP4 and VP7 forming the outer capsid. VP6 is formed in the middle capsid, VP1 and VP2 are formed the inner capsid. RV is classified into 11 serological groups (A-K) based on the antigenic properties of VP6 and described by Gx-P[x]-Ix-Rx-Cx-Mx-Ax-Nx-Tx-Ex-Hx (x indicating the numbers of the corresponding genotypes), representing the genotypes of VP7-VP4-VP6-VP1-VP2-VP3-NSP1-NSP2-NSP3-NSP4-NSP5, respectively [5]. With the latest statistics, at least 42 G genotypes, 58 P genotypes, 32 I genotypes, 28 R genotypes, 24 C genotypes, 24 M genotypes, 39 A genotypes, 28 N genotypes, 28 T genotypes, 32 E genotypes, and 28 H genotypes have been reported in humans and animal (https://rega.kuleuven.be/cev/viralmetagenomics/virus-classifica-tion/rcwg, accessed on 11 June 2024).

Rotavirus group A (RVA) was considered the most prevalent group for human beings and pig herds. In humans, G1P[8], G3P[8], G4P[8], G9P[8], G12P[8], and G2P[4] RVA strains were the main frequent genotypes worldwide. In pigs, G1-G6, G8-G12, and G26 along with P[1]-P[8], P[11], P[13], P[19], P[23], P[26], P[27], P[32], and P[34] were the predominant genotypes. The rearrangement and reassortment of genomic segments have been continuously reported in RVA. In fact, RVA strains are classified into three genogroups: Wa-like (genogroup 1), DS-1-like (genogroup 2), and AU-1-like (genogroup 3). Most human RVA (HuRVA) and porcine RVA (PoRVA) strains belong to the Wa-like genogroup with a Gx-P[8]-I1-R1-C1-M1-A1-N1-T1-E1-H1 genotype constellation, while bovine RVA (BoRVA) strains belong to the DS-1-like genogroup with a G2-P[4]-I2-R2-C2-M2-A2-N2-T2-E2-H2 genotype constellation. Other animal RVA strains belong to the AU-1-like genogroup with a G3-P[9]-I3-R3-C3-M3-A3-N3-T3-E3-H3 constellation [6]. Several studies on the reassortment or inter-species transmission of human and porcine RV strains are published annually. Examples include the GDJM1 strain (PoRVA) from China in 2022 [7], the CN127 strain (PoRVA) from China in 2020 [8], the SO1199 strain (HuRVA) from Japan in 2020 [9], and the KCH148 strain (HuRVA) from Kenya in 2019 [10].

The reassortment or inter-species transmission events have not typically involved normally sized gene fragments. Notably, the mutant NSP5 fragment, which is relatively conserved within specific species but often overlooked, has rarely been reported. To date, the super-short NSP5 gene has only been documented in Japan [11], Belgium [12], and Indonesia [13]. However, most NSP5 gene fragments are approximately 667 bp in length; fragments exceeding 667 bp are exceptionally rare, especially in China. Up to now, only a rabbit RV G3P[14] strains (N5) with 946 nucleotides for the NSP5 gene have been described in China [14]. RV finishes its replication in the viroplasms. Several viral proteins form the viroplasms; among these viral proteins, NSP5 acts as the main building block, hence, it was considered to be an essential component for RV replication [15]. NSP5 affects viroplasm formation and virus replication through interacting with NSP2, NSP6, and VP2 [16]. Therefore, it is of great significance to identify the PoRVA strains with special features in the NSP5 gene.

In this study, we firstly identified a special PoRVA strain with a duplicated sequence (344-nt) in the untranslated region (UTR) of the NSP5 gene (962-nt in length) during PoRVA surveillance in the Guangdong province of China. We aimed to characterise novel RV strains within porcine faecal samples and investigate the potential for cross-species transmission between pigs and humans.

## 2. Materials and Methods

### 2.1. Viral Genomic RNA Extraction from Faecal Samples

The two diarrhoeic faecal samples, named S2CF and S3CF, were collected from two 15-day-old nursing piglets admitted with acute diarrhoea at a pig farm located in Guangdong Province, China, in 2023 and stored at Animal Health Institute, Guangdong Academy of Agricultural Sciences. Total Viral RNA was extracted from 200 μL of the faecal samples using the RaPure Viral RNA/DNA Kit according to the manufacturer’s protocol (Guangzhou Magen Biotechnology Co., Ltd., Guangzhou, China). The total extracted RNA was stored at −80 °C until use.

### 2.2. Amplification and Sequencing of Rotavirus Genes

RVA detection was performed using a TaqMan probe-based real-time quantitative reverse transcription polymerase chain reaction (qRT-PCR) method developed in this study, targeting the NSP5 gene of RVA, which is more conserved than other fragments in rotaviruses of pigs, humans, cows, sheep, and horses. The VP6, VP7, and NSP1–NSP5 segments of PoRVA were amplified by one-step RT-PCR method (TaKaRa One Step PrimeScript^TM^ RT-PCR Kit, Otsu, Shiga, Japan). And the sequences of VP1 to VP4 segments were obtained with the infusion clone assay (TaKaRa In-Fusion Snap Assembly cloning kits, Otsu, Shiga, Japan). All primers and PCR thermal cycling conditions used in this study for 11 RVA segments’ amplification are detailed in Table 1. The basic PCR master mix details were same for amplifying all 11 segments. The final primer concentration was 0.4 μM, template volume was 50 μL, and thermal cycling was 40. The positive control was preserved in our laboratory. A DL2,000 DNA marker (TaKaRa, Dalian, China) was used in agarose gel electrophoresis. The PCR products from each gene fragment were purified using a Gel Extraction kit (Omega, Norwalk, CT, USA) according to the manufacturer’s instructions. The purified PCR products were cloned into the pMD19-T vector (TaKaRa, Dalian, China) and the clones were Sanger sequenced using M13 forward and M13 reverse primers. Sanger sequencing based on the chain Termination in triplicate was finished by a commercial research laboratory ((BGI Tech Solutions (Beijing Liuhe) Co., Ltd., Beijing, China). Finally, the complete gene sequences of RVA (n = 11) were achieved through sequencing amplicons from all RV segments and deposited in GenBank database.

### 2.3. Phylogenetic Recombination and Mutation Analyses

The reference sequences were downloaded in GenBank database, which meet the two following rules. Firstly, they are the top three sequences with the highest similarity when online blastn alignment was performed. Secondly, the genotype of the reference sequence should be as prevalent as possible. According to these rules, a total of 61 VP7, 63 VP4, 46 VP6, 41 VP1, 42 VP2, 34 VP3, 41 NSP1, 36 NSP2, 40 NSP3, 43 NSP4, and 45 NSP5 reference sequences were used to construct the evolutionary trees. The sequences used to construct the evolutionary trees were all trimmed to equal lengths post-alignment by MEGA version XI software.

The evolutionary trees were constructed based on each multiple sequence alignment obtained via Clustal W algorithm. The Maximum Likelihood (ML) and Neighbour-Joining (NJ) methods with 1000 bootstrap replicates from MEGA version XI software package were used to construct phylogenetic analyses of 11 segments [17]. The best-fit substitution models were selected based on the lowest Bayesian Information Criterion (BIC) scores as determined by the built-in model test application. To detect recombination, 11 aligned sequences of S2CF were analysed using RDP4 (version 4.101) with default settings across seven algorithms: RDP, GENECONV, Chimaera, MaxChi, BootScan, 3Seq, and SiScan. Recombination events were confirmed if at least four algorithms yielded significant p-values (*p* < 1.0 × 10^−6^) [18].

The nucleotide and protein sequences alignments analyses of the genome segments were performed using BLAST (Basic Local Alignment Search Tool) program from the National Centre for Biotechnology Information (NCBI, National Institutes of Health, Bethesda, MD, USA). The nucleotide similarities were conducted via Megalign module of the DNASTAR Lasergene.v7.1 software package (DNASTAR, Inc., Madison, WI, USA). Amino acid alignments and mutation analyses were constructed with MEGA version XI software package.

### 2.4. Assignment of Genotypes and Accession Numbers

The genotype of each gene was determined via combining the classification tool for group A rotaviruses (RotaC v2.0, http://rotac.regatools.be/, accessed on 11 June 2024), proposed by the Rotavirus Classification Working Group (RCWG), with phylogenetic trees. The eleven nucleotide sequences have been deposited in the GenBank database (https://www.ncbi.nlm.nih.gov/, accessed on 11 June 2024) and accession numbers are showed in Data Availability Statement.

## 3. Results

### 3.1. RT-PCR and Agarose Gel Electrophoresis Analyses

For the positive faecal sample, nearly full-length sequences of 11 segments were successfully obtained via RT-PCR and visualised by agarose gel electrophoresis. Unexpectedly, a slightly larger electrophoretic band (about 1000 bp) than the expected size (667 bp) was observed in the NSP5 gene of strain RVA/Pig-tc/CHN/S2CF/2023/G4P[6], abbreviated as S2CF (Figure 1), while the NSP5 gene of strain S3CF was about 667 bp in length. Additionally, the electrophoretic band sizes of the other 10 genes met normal expectations.

### 3.2. Whole Genome Acquirement and Similarity Analyses

We obtained nearly full-length gene sequences of the strain S2CF (GenBank no. PP255809-PP255819) and further characterised their similarity with reference strains from the GenBank database. As showed in Table 2, the VP7 gene is 1062 bp in length and shares the highest nucleotide sequence identity (96.07%) with the Chinese strain HLJ/15/1 from Heilongjiang. The VP4 (2359 bp) and VP6 (1356 bp) genes exhibit maximum nucleotide sequence identity of 96.40% and 95.87% with the Chinese porcine-like human strain E931 (G4P[6]), respectively [19]. The VP1, VP2, and VP3 genes were 3302 bp, 2717 bp, and 2527 bp in length, respectively, showing maximum nucleotide sequence identity (95.00%, 92.01%, and 96.87%) with the Chinese porcine strain YT(G4P[7]) and HeNNY-01(G4P[23]), and a human strain R479 (G4P[6]) [20]. For non-structure proteins, the NSP1, NSP2, NSP3, and NSP4 genes of strain S2CF exhibited the highest nucleotide sequence identity (96.03%, 95.50%, 95.63%, and 95.47%) with those of porcine and human strains, SC11 [21], TWN/4-1 [22], LL4260, and CN127 [8], from Mainland China and Taiwan, China, respectively. Interestingly, the NSP5 gene fragment is quite unique due to its additional 344-nt duplication insertion (Figure 2). Most NSP5 genes in length are of <670 bp, while that of the S2CF strain is of 962 bp in length and showed the maximum nucleotide sequence identity (91.90%) with that of the Belgium porcine-like human G9P[6] RV strain (BE2001) isolated from an infant [10].

The nearly complete sequence of the NSP5 gene was obtained through Sanger sequencing characterised as an additional 344 bases from the 619th to the 962nd nucleotide at the 3′ UTR end of the NSP5 gene when aligned with Wa (USA, Human RVA), Gottfried (USA, Porcine RVA), DU2014-259 (Thailand, Human RVA), and CMH-N016-10 (Thailand, Human RVA) (Appendix A). Using the Gottfried strain as a reference sequence, the 344 duplicated nucleotides start at the 307th nucleotide site and end at the 664th nucleotide site (Figure 2). The raw Sanger sequencing data and sequence alignments are provided in Appendix A, respectively.

### 3.3. Phylogenetic and Recombination Relationships of All Genes

Each of the 11 gene segments of the strain S2CF were characterised via phylogenetic trees based on the nearly full-length gene sequences. All reference sequences were obtained from the GenBank database.

For the outer structure proteins, the VP7 and VP4 proteins were the main neutralisation antigens, the VP7 gene was assigned to the G4 genotype and clustered with two porcine-derived strains, HLJkd and HLJ/15/1, from China and one porcine-derived strain, MRC-DPRU1557, from South Africa, respectively (Figure 3). And S2CF shared an 84.9–96.2% nucleotide identity with other G4 strains. Concerning the VP4 gene, the identity ranged from 84.3% to 96.6%. The S2CF strain clustered with the G4P[6] human strains E931, R1954, GX54, LL3354, and R479 from China and the strains CMH-N016-10 and CMH-N014-11 from Thailand (Figure 4). For another outer-structure protein, the phylogenetic analysis of the S2CF VP6 gene indicated that it belonged to group I1 among the 13 groups (Appendix A). The S2CF strain revealed a 91.4–95.9% identity with other group A rotaviruses clustered in the same group.

For the inner structural proteins, the analysis of the VP1 gene revealed a sequence identity ranging from 85.4% to 95%. The phylogenetic analysis of VP1 indicated that the strain S2CF formed a distinct cluster alongside several Chinese porcine isolates (the R1 clade), such as JS, YT, and CN127, distinct from bovine, human, feline, and pigeon RVs (Appendix A). Interestingly, strains JS (G5P[23]) and CN127 (G12P[7]) represent the reassortant RVs between the porcine and human strains, with a Wa-like backbone. In addition, the evolutionary relationship between the strains S2CF and YT is close, but we did not find any evidence to prove whether the YT strain is also a human-like porcine recombinant RV. Nevertheless, it does not affect the speculation that the S2CF strain is also a human-like porcine reassortment RV. The VP2 gene of the S2CF strain clustered in C1 genotype and shared 86.9–95.5% nucleotide (nt) similarity with other strains. As we can see, the VP2 gene revealed that S2CF grouped within the C1 genotype and was closest to a human RV, R1207, detected in Sri Lanka with 100 bootstrap reliabilities. In addition, the VP2 gene also clustered with some human and human–porcine origin strains from China (Appendix A). In contrast to the VP1 and VP2 genes, the VP3 segment exhibited a nucleotide identity of 85.0–96.9% with the other M1 strains. The VP3 seemed likely to be genetically more heterogenous for clustering within porcine, porcine–human, human, and giant panda strains from China (Appendix A).

In terms of non-structure proteins, the NSP1 genes clustered in genotype A8 together with porcine strains purely with identity of 80.2–96.5%. They appeared to be distant from other human and porcine strains (Appendix A). The NSP2 genes of the S2CF strains fell into Gottfried representative clusters with identity of 87.3–95.5%. The closest strain of genetic distance is a G9 strain, 4-1 isolated in Taiwan, China, which formed a cluster with other porcine and human isolates from China, Thailand, the USA, and Belgium with a 95.5% nucleotide identity. This segment was more closely related to porcine strains than to typical human strains (Appendix A). The NSP3 genes, as shown in Appendix A, formed a single lineage that is neighbouring to Chinese porcine HeNNY-01 and HLJ/15/1 strains and a Rabbit strain Z3171, isolated in 2020 from Shandong, China [23]. The identities of this segment were 89.0–96.4% compared with all T1 strains. Similarly to most porcine rotaviruses, the NSP4 gene segment of S2CF was assigned to genotype E1 which is a major clade containing Chinese porcine strains and was closely related to HeNNY-01 (Appendix A). The NSP4 sequence has an 89.9–95.5% identity to all E1 strains from China, the USA, and Thailand.

The NSP5 gene of the S2CF strains belonged to the H1 genotype, formed a separate clade, and clustered with many human rotaviruses detected in Belgium, Thailand, the USA, and China (Figure 5). The analysis of the NSP5 gene revealed sequence identity ranging from 69.4% to 98.6%.

The intragenotype or intergenotype recombination events were not detected in all 11 gene segments with the RDP4 recombination analysis [22]. Herein, the 11 segments of the S2CF strain were not a recombination strain.

### 3.4. The VP7 and VP4 Antigenic Region Analyses

The VP7 gene was shown to contain three antigenic epitope regions, 7-1a, 7-1b, and 7-2, which were made up of 29 amino acid residues [24]. The VP7 epitopes of the S2CF strain showed 28 amino acid differences compared with the Wa strains, and only one residue in position 212 of the 7-1b region was conserved. But, it presented an 89.66–100% identity compared with the same cluster strains without a unique mutation (Table 3), as far as the antigenic epitope regions of the VP4 gene, which consist of 25 amino acid residues in the 8-1 to 8-4 antigenic epitope regions. They shared a 96% to 100% identity. And the 12 amino acid residues in the 5-1 to 5-5 antigenic epitope regions exhibited the greatest similarities without any different amino acid residues (Table 4). All amino acid residues located in the binding region of the monoclonal antibody sites were conserved.

### 3.5. Genotyping

According to the latest classification and naming system established by the RCWG, the nucleotide identity cutoff values for 11 gene segments of Rotavirus Group A are as follows: 80% (G), 80% (P), 85% (I), 83% (R), 84% (C), 81% (M), 79% (A), 85% (N), 85% (T), 85% (E), and 91% (H) [22]. The genotype of S2CF was identified as G4-P[6]-I1-R1-C1-M1-A8-N1-T1-E1-H1.

## 4. Discussion

RV infections pose significant challenges to human as well as animal health throughout the world, particularly in developing countries that have faced substantial losses due to the virus [25]. Most children under 5 years old across the world have encountered acute diarrhoea and gastroenteritis caused by RV [26,27,28,29]. More than 230,000 deaths are attributed to RV each year globally [30]. It cannot be ignored that, as a major veterinary pathogen, RV significantly affects agricultural production, mainly involving weaning and post-weaning piglets, young calves, young sheep/goats as well as foals [31]. In a pig farm from the Guangdong Province of China, according to one previous study, the morbidity and mortality of RV were 60.00% and 20.99% in 2022, respectively [7].

RV has 11 serogroups and lots of genotypes. Due to its diversity and mutations, prevention and control is difficult. Frequent contact exists between humans and pigs, which causes an increasing possibility of the cross-species transmission of RV. The RVA porcine-to-human reassortment events usually occur in Africa [32,33,34,35,36], Asia [37,38,39,40], Europe [41,42,43], and America [44]. The genotypic analysis of RV is beneficial for understanding the presence of RV between humans and pigs. Among them, the G4P[6] stains were the most prevalent genotype in porcine-to-human or human-to-porcine RV. During 2000 and 2004, the BP271 and BP1125 strains were collected in Hungary [42]; NT0042 and NT0077 were isolated from June 2007 to August 2008 in Vietnam [45]; the strains E931 from 2008 and R1954 from 2013 were detected in China, respectively [19]; the PZ3 strain was obtained from Italy in 2017 [46]; DU2014-259 and PK2015-1-0001 appeared in Thailand, in 2021 [40]; and so on, which were all characterised as the porcine-derived human RVA strains detected from children with acute gastroenteritis and analysed to be the porcine-to-human interspecies transmission events. However, in this study, the S2CF strain constellated a G4-P[6]-I1-R1-C1-M1-A8-N1-T1-E1-H1 genotype configuration, which shows that it was likely that S2CF emerged as a human-to-porcine interspecies strain. The reasons were summarised as (i) its origin was in faecal samples collected in piglets, (ii) porcine RV strains occupy a typical A8 genotype for NSP1 segments, as do the S2CF strain, and (iii) nearly half of the gene segments, including VP4 (P[6]), VP6 (I1), VP3 (M1), NSP3 (T1), and NSP5 (H1), exhibited the highest homology ranging from 91.90–96.87% similarity to human RVA strains, while another six segments, including VP7 (G4), VP1 (R1), VP2 (C1), NSP1 (A8), NSP2 (N1), and NSP4 (E1), of this strain were the most closely related to the porcine strains, with the highest homology ranging from 92.01–96.07% similarity.

For RV, NSP5 is a unique segment encoding two open reading frames which contain an internal ORF for NSP6 [47]. We suggest that the extension of the NSP5 gene has little effect on the VP7 and VP4 antigenic epitopes because the antigenic epitopes of VP7 and VP4 did not change significantly compared with the strains from the same clusters. So, the ability of antibodies to neutralise virus infectivity and vaccine effectiveness was not obviously affected [20]. But importantly, NSP5 can help RV escape against natural immune surveillance by forming the viroplasm with NSP2 in order to promote RV replication [45,48,49]. Therefore, it cannot be ruled out that the changes of the NSP5 gene in length will affect the replication and even evolution of RV. Throughout the entire analysis of the 11 segments, the NSP5 gene fragment should be concerned mostly for its significant variability. In the present study, for the strain of S2CF, the NSP5 gene was 962 bp in length, which is longer than most rotaviruses. Compared to other normal NSP5 fragments of RVs, S2CF has 344 additional nucleotides at 3′ UTR. To our knowledge, the duplicated sequence in the NSP5 gene has not been reported in a Chinese porcine or human RV before. But, some duplicated sequences in the NSP5 gene have rarely been described in Argentina (porcine RV) [50], Japan (AU-19, human RV) [11], and Indonesia (57M, 69M, human RV) [13]. According to our records, when the whole 11 segments of RVs were not carried out before 2008, abnormal NSP5 fragments were only found by the polyacrylamide gel electrophoresis (PAGE) of faecal or tissue culture samples [51]. In these special strains detailed gene sequence information was absent, so we cannot know which region of the abnormal fragment has experienced an abnormality. After the promotion of sequencing the full length of 11 segments, it is helpful for us to gain a clear understanding of the abnormal information of abnormal fragments. For example, B4106 was isolated from a child with severe gastroenteritis in Belgium [52] and the NSP5 gene had 1043 nucleotides. And the NSP4 gene of B4106 was grouped in the lapine RV including the ALA, BAP, and BAP-2 strains, which indicated the cross-species transmission attribute between sheep and humans. This does not come singly but in pairs, as BE2001 was also discovered in Belgium with rearranged NSP5 fragments. The difference is that the BE2001 strain was a porcine-like human RV with the 300-nucleotides duplication in the 3′ end of NSP5 [12]. Similarly, AU19 was a reassortment human strain from porcine with the 949 nucleotides of the NSP5 gene in 2014, Japan [11].

It was notable that these strains with abnormal genome patterns were all from children, but in this study, we monitored the strain of S2CF which is a porcine-origin strain containing some genes which are similar to human and porcine rotaviruses with a rearranged NSP5 gene. According to previous studies, before the 1990s, rotaviruses with genome rearrangements existed in immunocompetent hosts including calves [53], pigs [50], lapine [54], and humans [55,56]. However, for nearly 20 years, NSP5 has been manipulated to express an additional gene via the reverse genetics system [47,57]. Hence, it is a puzzle to understand where it is from or how it was formed, but we think this may be a cross-host transmission event from humans to pigs. Therefore, it is very important to pay attention to the occurrence and development of RV in humans and pigs. Regretfully, due to various reasons, we were unable to collect human faeces from this pig farm for the monitoring and analysis of RV. But this is also a very important issue, so in the future, we propose that the monitoring and analysis of porcine RV should be synchronised with the monitoring of human faeces in pig farms. The analysis of the correlation between pigs and humans will obtain a better understanding of the epidemic trend, occurrence, and development mechanism of RV.

Based on the whole genome analyses of RV, it is helpful to find the derivation and evolutionary patterns of RV. Due to a single nucleotide mismatch in the NSP5 gene, the strain of Mc323, lacking the putative ORF encoding NSP6, was reported. And due to a partial duplication in the 3′ UTR of NSP5, the strain of Mc345 was found in Thailand [58]. The HuRVA strain 69M has a low degree of genetic homology and a 20-fold difference in neutralisation titers, which suggests that a new serotype was first reported in Japan, 1985 [59]. But in 2008, the full-genome sequence of 69M was actually presented and regarded as a new genotype DS-1 [60]. Similarly, NSP6 ORFs of five HuRVAs (D, P, WI61, AU-1, and N26-02) were described as truncated and two HuRVAs (Se584 and A64) were extended when the full-genome sequences were provided [60]. Therefore, whole-genome analyses can help to discover the diversity of RV. But in China, a lot of studies have only focused on the identification of the VP7 and VP4 genes of RVA, and little is available about the diversity of other genes, especially the NSP5 gene. This may be the reason why there is no report on the occurrence of a duplication sequence in the NSP5 gene for porcine RV in China.

Generally speaking, the RVA strains with the duplicated sequence insertions in the NSP5 gene are rare in the world. H1, H2, and H3 out of 28 H genotypes were reported with the duplication of normal NSP5 [61]. However, to our knowledge, our description of the duplication in segment 11 is the first description in pig RV strains in China. The resolution of the phylogenetic relationship between porcine and human RVA genes was occasionally low, and sometimes it made the evaluation of the host species origin of individual genes difficult [42]. Moreover, further research about each gene segment of RV, including the NSP5 gene fragment, is necessary to evaluate whether the insertion or deletion affects the cross-species transmission and pathogenic mechanism of RV. Consequently, it is critical to obtain the RV full genome for elucidating the patterns of virus evolution [14].

## 5. Conclusions

Taken together, we identified a unique PoRVA strain with a longer NSP5 gene from a piglet with diarrhoea in the Guangdong Province, China. The genome analysis suggests that inter-species transmission between porcine and human RVAs possibly occurred in China. Importantly, the newly emerging longer NSP5 gene indicates that RVA becomes more and more complex in China. Due to the zoonotic ability, based on the concept of “one world, one health”, the 11 segments’ genomic analyses of RV are warranted, which will provide important insights into inter-species transmission and the origin of RV.

## Figures and Tables

**Figure 1 animals-14-01790-f001:**
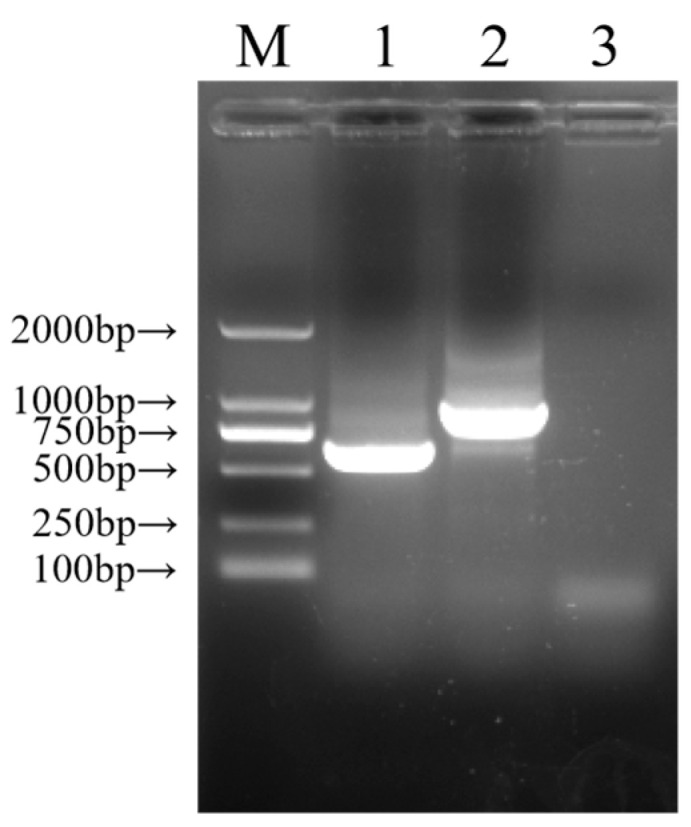
Visualised agarose gel electrophoretic diagram of NSP5 segments. M, DL2,000 DNA marker; 1, positive control; 2, S2CF strain; 3, negative control.

**Figure 2 animals-14-01790-f002:**
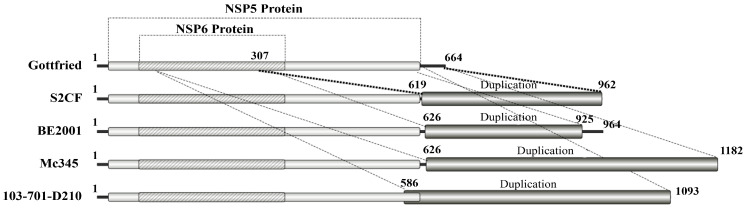
The schematic representation of the duplication events. The duplicated sequence in each of the strains was shown in dark grey.

**Figure 3 animals-14-01790-f003:**
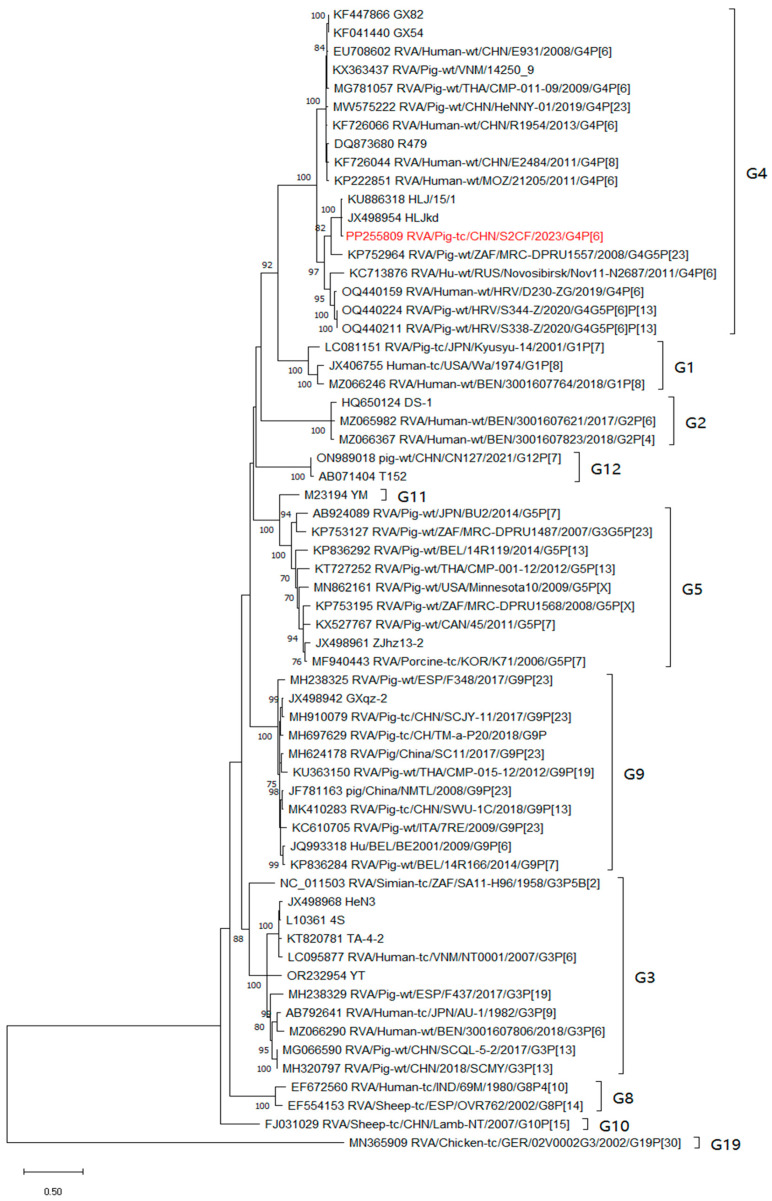
Maximum likelihood phylogenetic tree based on VP7 coding sequences, calculated with 1000 bootstrap replicates under the GTR + G + I model. Node labels representing bootstrap values ≥ 70% are shown. The S2CF strain isolated in this study is marked with red. Abbreviation: GTR = General Time Reversible, G = Gamme sites, I = Invariant sites.

**Figure 4 animals-14-01790-f004:**
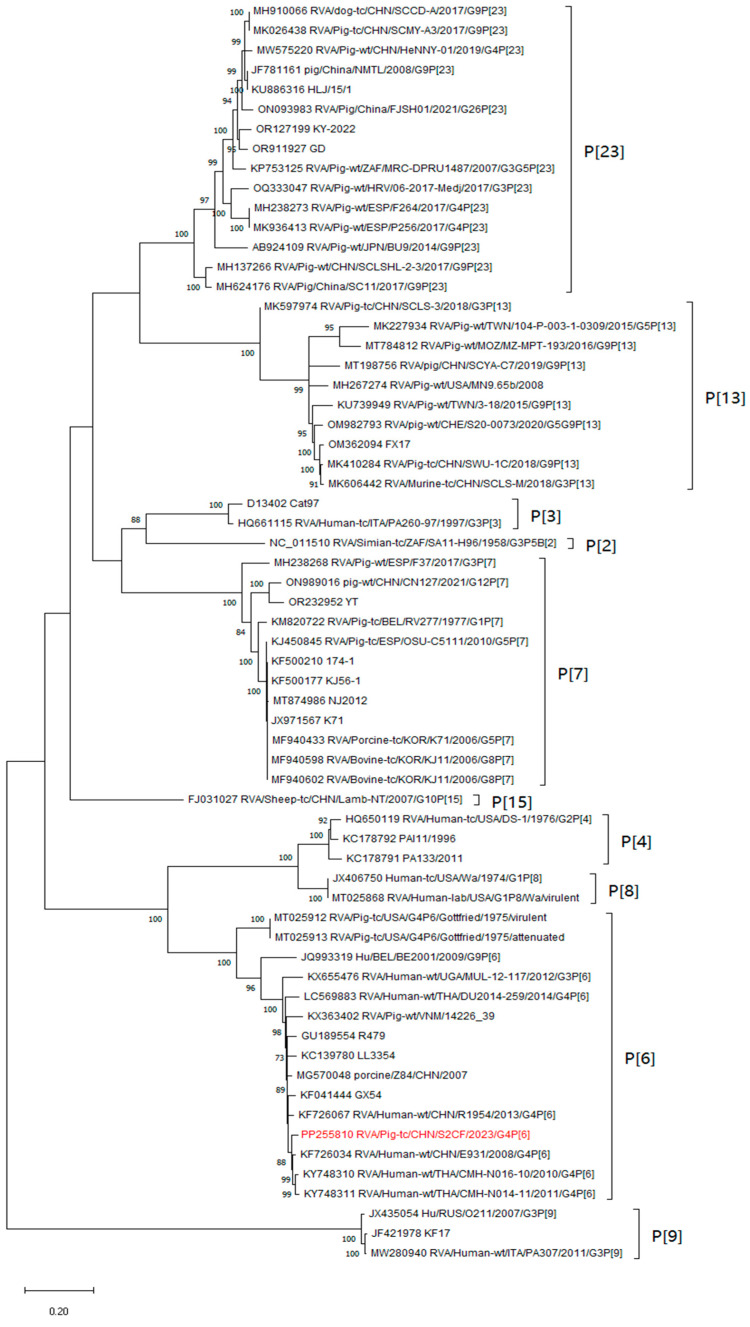
Phylogenetic dendrograms of the segment encoding VP4 was estimated using genetic distances calculated by maximum likelihood with 1000 bootstrap replicates under the GTR + G + I model. The bootstrap values ≥ 70% are indicated at each branch node represents substitutions per nucleotide site. The S2CF strain isolated in this study was marked in red.

**Figure 5 animals-14-01790-f005:**
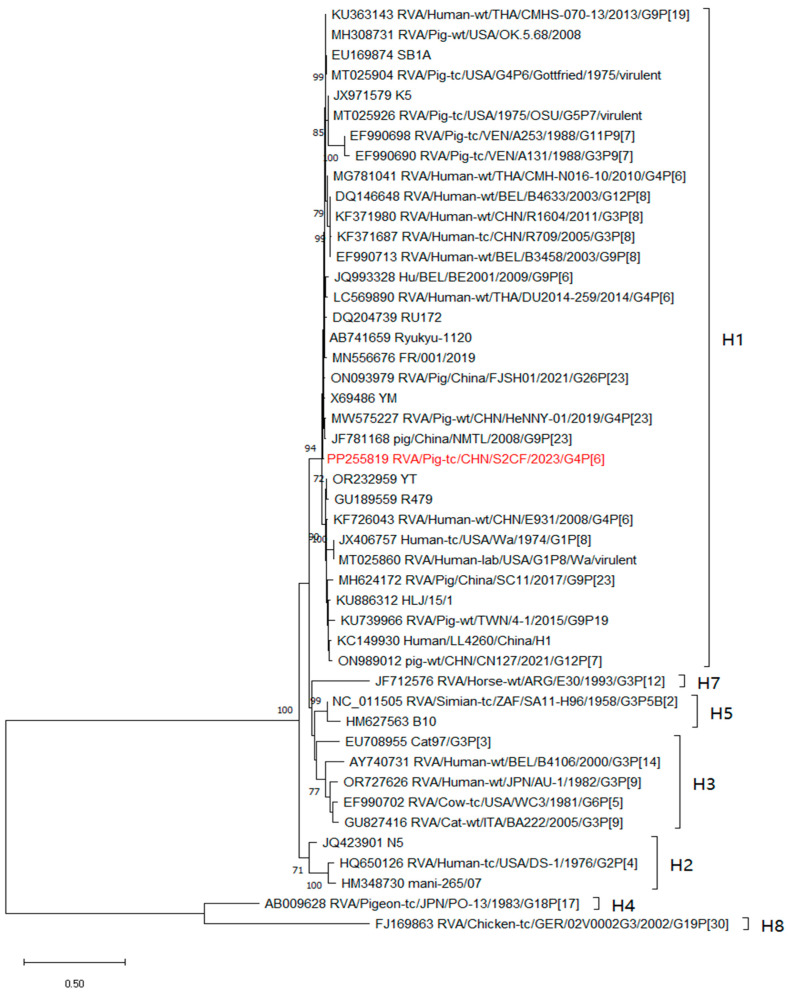
Phylogenetic tree constructed based on the segments of NSP5 was constructed using neighbour-joining method with 1000 bootstrap replicates. Node labels representing bootstrap values ≥ 70% are shown. The S2CF strain isolated in this study is marked with red.

**Table 1 animals-14-01790-t001:** List of primers for each segment of RVA amplification used in RT-PCR.

Name	Sequence (5′–3′)	PCR Product Size (bp)	Annealing Temperature (°C)	Extension Time
VP1	F^1^: CGGTACCCGGGGATCGGCTATTAAAGCTGTACAATG	3302	47.8	3 min 25 s
	R^2^: CGACTCTAGAGGATCGGTCACATCTAAGCRCTCTAATC			
VP2	F: CGGTACCCGGGGATCGGCTATTAAAGGYTCAATGGC	2690	47.8	2 min 45 s
	R: CGACTCTAGAGGATCGGTCATATCTCCACARTGGG			
VP3	F: CGGTACCCGGGGATCATGAAAGTATTAGCTTTAAGAGACA	2507	64.4	2 min 45 s
	R: CGACTCTAGAGGATCTCACTCAGRCATATCAAATACGG			
VP4	F: ATGGCTTCGCTCATTTACAGACAATTG	2360	56.8	2 min 35 s
	R: GGTCACAACCTCTCGACATTGCTTACAGT			
VP6	F: GGCTTTWAAACGAAGTCTTCGACAT	1356	52.5	1 min 30 s
	R: GGTCACATCCTCTCACTAYAYCAT			
VP7	F: GGCTTTAAAAGAGAGAATTTC	1062	48.0	1 min 10 s
	R: GGTCACATCAWACARTTCTA			
NSP1	F: TATGAAAAGTCTTGTGGAAGCCAT	1550	50.6	1 min 45 s
	R: ATGCTGCCTAGGCGCTACTCTAGTGCA			
NSP2	F: GGCTTTTAAAGCGTCTCAGT	1059	52.5	1 min 10 s
	R: GGTCACATAAGCGCTTTCTATTCT			
NSP3	F: CAGTGGTTGATGCTCAAGATGGAG	1020	56.3	1 min 10 s
	R: CTATTGTGCTCATAGAGGGTCATGTGTA			
NSP4	F: GGCTTTTAAAAGTTCTGTTCCG	749	57.5	50 s
	R: GGTCACACTARGACCATTCCTTCCAT			
NSP5	F: GGCTTTAAAAGCGCTACAGTGATGT	667	64.4	45 s
	R: GAGTGGGGAGCTCCCTAGTGTGCTCT			

F^1^: forward primers, R^2^: reverse primers.

**Table 2 animals-14-01790-t002:** Comparative analysis of nucleotide similarity in 11 segments of S2CF strain.

Gene	Length(bp)	Closely Related Strains	Country	Accession No.	nt (%)
VP7	1069	HLJ/15/1	China	KU886318	96.07
VP4	2359	RVA/Human-wt/CHN/E931/2008/G4P[6]	China	KF726034	96.40
VP6	1356	RVA/Human-wt/CHN/E931/2008/G4P[6]	China	KF726035	95.87
VP1	3302	YT	China	OR232949	95.00
VP2	2717	RVA/Pig-wt/CHN/HeNNY-01/2019/G4P[23]	China	MW575218	92.01
VP3	2527	RVA/Human-wt/CHN/R479/2004/G4P[6]	China	GU189553	96.87
NSP1	1528	RVA/Pig/China/SC11/2017/G9P[23]	China	MH624168	96.03
NSP2	1044	RVA/Pig-wt/TWN/4-1/2015/G9P[19]	China: Taiwan	KU739963	95.50
NSP3	1007	Human/LL4260/China/T1	China	KC149929	95.63
NSP4	750	pig-wt/CHN/CN127/2021/G12P[7]	China	ON989011	95.47
NSP5	962	Hu/BEL/BE2001/2009/G9P[6]	Belgium	JQ993328	91.90

**Table 3 animals-14-01790-t003:** The alignment of the amino acid residues in VP7 (7-1a, 7-1b, and 7-2) antigenic epitopes.

Accession Number	Strains	7-1a														7-1b						7-2								
		87	91	94	96	97	98	99	100	104	123	125	129	130	291	201	211	212	213	238	242	143	145	146	147	148	190	217	221	264
AB792641	AU-1	T	T	N	N	S	W	K	D	Q	D	A	V	D	K	Q	D	T	N	N	N	K	D	A	A	L	S	E	A	G
HQ650124	DS-1	A	N	S	D	E	W	E	N	Q	D	N	V	N	K	Q	D	V	N	N	N	R	D	N	T	S	D	I	S	G
K02033	Wa	V	E	N	V	D	F	S	V	L	M	E	L	I	N	V	T	T	T	N	I	L	Y	Y	Q	Q	A	I	K	W
KF447866	GX82	T	T	S	N	E	W	K	D	Q	N	L	V	D	K	Q	N	T	N	D	T	R	V	S	G	E	S	T	S	G
KF041440	GX54	T	T	S	N	E	W	K	D	Q	N	L	V	D	K	Q	N	T	N	D	T	R	V	S	G	E	S	T	S	G
EU708602	E931	T	T	S	N	E	W	K	D	Q	N	L	V	D	K	Q	N	T	N	D	T	R	A	S	G	E	S	T	S	G
KX363437	14250_9	T	T	S	N	E	W	K	D	Q	N	L	V	D	K	Q	N	T	N	D	T	R	A	S	G	E	S	T	S	G
MG781057	CMP-011-09	T	T	N	N	E	W	K	D	Q	N	L	V	D	K	Q	N	T	N	N	T	R	I	S	G	E	L	T	S	G
MW575222	HeNNY-01	T	T	N	N	E	W	K	D	Q	N	L	V	D	K	Q	N	T	N	D	T	R	A	S	G	E	S	T	S	G
KF726066	R1954	T	T	N	N	E	W	K	D	Q	N	L	V	D	K	Q	N	T	N	D	T	R	A	S	G	E	S	T	S	G
DQ873680	R479	T	T	S	N	E	W	K	D	Q	N	L	V	D	K	Q	N	T	N	D	T	R	A	S	G	E	S	T	G	G
KF726044	E2484	T	T	S	N	E	W	K	D	Q	N	L	I	D	K	Q	N	A	N	D	T	R	A	S	G	E	S	T	S	G
KP222851	21250	T	T	S	N	E	W	K	D	Q	N	L	V	D	K	Q	N	T	N	D	T	R	A	S	G	E	S	T	S	G
KU886318	HLJ/15/1	T	T	G	N	E	W	K	D	Q	N	L	I	D	R	Q	N	T	G	D	T	R	I	A	G	E	S	M	N	G
JX498954	HLJkd	T	T	S	N	E	W	K	D	Q	N	L	I	D	R	Q	N	T	G	D	T	R	I	A	G	E	S	M	N	G
KP752964	DPRU1557	T	T	G	N	E	W	K	D	Q	N	L	I	D	R	Q	N	A	G	D	T	R	I	A	G	Q	S	T	N	G
KC713876	NoV11-N2687	A	T	S	N	E	W	K	D	Q	N	L	V	D	R	Q	N	T	G	D	T	R	V	A	G	E	S	M	N	G
OQ440159	D230-ZG	A	T	G	N	E	W	K	D	Q	N	L	I	D	R	Q	N	A	G	D	T	R	I	A	G	E	S	M	N	G
OQ440224	S344-Z	A	T	G	N	E	W	K	D	Q	N	L	I	D	R	Q	N	A	G	D	T	R	I	A	G	E	S	M	N	G
OQ440211	S338-Z	A	T	G	N	E	W	K	D	Q	N	L	I	D	R	Q	N	A	G	D	T	R	I	A	G	E	S	M	N	G
PP255809	S2CF	A	T	G	N	E	W	K	D	Q	N	L	I	D	R	Q	N	T	G	D	T	R	I	A	G	E	S	M	N	G

The grey background represents amino acid position of residues associated with escape neutralisation with monoclonal antibodies. Different amino acid residues of S2CF were rendered in red, compared with the Wa strain.

**Table 4 animals-14-01790-t004:** The alignment of the amino acid residues in VP4 (8-1 to 8-4 and 5-1 to 5-5) antigenic epitopes.

Accession Number	Strains	8-1											8-2		8-3									8-4			5-1								5-2	5-3	5-4	5-5
		100	146	148	150	188	190	192	193	194	195	196	180	183	113	114	115	116	125	131	132	133	135	87	88	89	384	386	388	393	394	398	440	441	434	459	429	306
D10970	AU-1	D	L	P	G	Y	L	I	N	N	D	N	A	N	Q	N	T	Q	N	S	N	D	S	T	R	E	S	S	A	N	H	S	S	R	E	A	R	T
HQ650119	DS-1	D	S	Q	D	S	T	D	L	N	N	I	T	A	S	Q	T	N	N	E	N	N	D	N	T	N	Y	F	L	W	P	G	R	T	P	E	L	R
L34161	Wa	D	S	Q	E	S	T	N	L	N	N	I	T	A	N	P	V	D	S	S	N	D	N	N	T	N	Y	F	I	W	P	G	R	T	P	E	L	R
MT025912	Gottfried-V	D	N	N	D	S	T	N	L	P	D	V	T	A	P	S	Q	D	V	E	N	S	D	I	N	K	Y	F	I	W	P	G	R	T	P	E	L	R
MT025913	Gottfried-A	D	N	N	D	S	T	N	L	P	D	V	T	A	P	S	Q	D	V	E	N	S	D	I	N	K	Y	F	I	W	P	G	R	T	P	E	L	R
JQ993319	BE2001	D	N	S	E	S	T	N	L	P	D	I	T	A	T	S	Q	D	T	E	N	N	S	T	N	Q	Y	F	I	W	P	G	R	T	P	E	L	R
KX655476	MUL-12-117	D	S	S	E	S	T	N	L	S	E	V	T	A	T	N	Q	R	T	E	N	N	N	T	N	Q	Y	F	I	W	P	G	R	T	P	E	L	R
LC569883	DU2014-259	D	S	N	E	S	T	N	L	S	E	I	T	A	A	N	G	N	T	E	N	N	N	T	N	Q	Y	F	I	W	P	G	R	T	P	E	L	R
KX363402	14226_39	D	S	S	E	S	T	N	L	S	E	V	T	A	T	N	Q	S	T	E	N	N	N	T	N	Q	Y	F	I	W	P	G	R	T	P	E	L	R
GU189554	R479	D	S	S	E	S	T	N	L	S	E	V	T	A	T	N	Q	S	T	E	N	S	N	T	N	Q	Y	F	I	W	P	G	R	T	P	E	L	R
KC139780	LL3354	D	S	S	E	S	T	N	L	S	E	V	T	A	T	N	Q	S	T	E	N	N	N	T	N	Q	Y	F	I	W	P	G	R	T	P	E	L	R
MG570048	Z84	D	S	S	E	S	T	N	L	S	E	V	T	A	T	N	Q	S	T	E	N	N	N	T	N	Q	Y	F	I	W	P	G	R	T	P	E	L	R
KF041444	GX54	D	S	S	E	S	T	N	L	L	E	V	T	A	T	N	Q	S	T	E	N	N	N	T	N	Q	Y	F	I	W	P	G	R	T	P	E	L	R
KF726067	R1954	D	N	S	E	S	T	N	L	S	E	V	T	A	T	N	Q	S	T	E	N	N	N	T	N	Q	Y	F	I	W	P	G	R	T	P	E	L	R
KF726034	E931	D	S	S	E	S	T	N	L	S	E	V	T	A	T	N	Q	S	T	E	N	N	N	T	N	Q	Y	F	I	W	P	G	R	T	P	E	L	R
KY748310	CMH-N016-10	D	S	S	E	S	T	N	L	S	E	I	T	A	T	N	Q	S	T	E	N	D	N	T	N	Q	Y	F	I	W	P	G	R	T	P	E	L	R
KY748311	CMH-N014-11	D	S	S	E	S	T	N	L	S	E	V	T	A	T	N	Q	S	T	E	N	N	N	T	N	Q	Y	F	I	W	P	G	R	T	P	E	L	R
PP255810	S2CF	D	S	S	E	S	T	N	L	S	E	V	T	A	T	N	Q	S	T	E	N	N	N	T	N	Q	Y	F	I	W	P	G	R	T	P	E	L	R

The grey background represents amino acid position of residues associated with escape neutralisation with monoclonal antibodies. Different amino acid residues of S2CF were rendered in red, compared with the Wa strain.

## Data Availability

The GenBank accession numbers for the 11 segments of S2CF strain are PP255809 to PP255819.

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
