# Peer review of "Emergence of a Novel G4P[6] Porcine Rotavirus with Unique Sequence Duplication in NSP5 Gene in China"

_animals, 2024, doi:10.3390/ani14121790_

Round 1

Reviewer 1 Report

Comments and Suggestions for Authors

This manuscript describes a genomic characterization of a rotavirus from a pig. Notably, the NSP5 gene has a rare duplication, and the frequent clustering of pig and human RVs in phylogenetic analyses suggests there may be frequent cross-species transmission. I think the study is important in advancing “One Health,” but I think the motives/justification for the study should be more clearly stated. It’s unclear if the point was to look for a mutant NSP5 and why this would be important.

Introduction

Line 49-52: This sentence explaining RV grouping is a bit difficult to understand at first. Is it correct that there are two different classification methods, one based only on V6 (A-L), and the other which considers genotypes for all the proteins? Is it correct that “x” is just a number that denotes a specific genotype of all those that are known (not that it indicates the number of genotypes present in a strain)? Why is “x” in brackets for VP4 (P[x]), but not the others? Maybe a supplemental table or figure could help illustrate this.

Line 57: What defines rotavirus group A? Are there any specific VP6 mutations that would result in a rotavirus being classified this way?

Line 58-60: Given the explanation of genotype notation in the previous paragraph, I’m a little confused here. In humans, how could G1, G3, G4, G9, and G12 be combined with G2P[4]? Are there duplications of the VP7 gene that could result in two different G genotypes being present? I think in this section you should specify you are talking about the most frequent VP7 (G) and VP4 (P) genotypes for clarity.

Line 74-76: Can you clarify what you mean by the “mutant NSP5 fragment”? Is it rare to have this fragment at all, or simply rare to observe a certain mutant version of that fragment? If the latter, what is considered to be the “mutant” version, and what is the wild type? Can you also clarify what you mean by “super short”? How much shorter is it than the other NSP genes?

Line 77-78: Please provide a citation for the 666 bp length mentioned here; it seems like an oddly specific threshold. (This exact number also seems to change throughout the paper, as I note below, so this value needs to be confirmed.)

Line 78-79: How does this rabbit rotavirus relate to length, as you were just discussing? Why is the length of this gene important for the virus and important to study?

Line 80: Has a similar repeat sequence ever been found in other rotaviruses? Why did you decide to look in pigs specifically?

Methods

Line 85: Please list the total number of samples used.

Line 95: Is this referring to the In-Fusion® clone assay from Takara Biosciences? Please specify.

Table 1: Does the “Length” column refer to the length of the expected product, or length of what you actually found? In the bottom of this column, why is the length for NSP5 given as 667 bp, while you stated in the introduction that fragments over 666 bp are extremely rare?

Line 116: This sentence is missing what specifically was performed with BLAST for nucleotide and protein sequences. Was this for taxonomic assignment?

Line 120: What types of mutation analyses were conducted? Does this just refer to finding mutations within an alignment?

Results

Line 131: There was one positive sample out of how many total samples? Is there any specific background for this individual that you could include (for example, were there any symptoms)?

Line 132-134: Please clarify that this band is for NSP5 specifically. Here, 651 bp is stated as the expected size, but previously it is given as 666 bp and 667 bp. Please address this inconsistency. Can you also clarify if the S2CF strain is a novel strain you identified and named, or if this has been found previously?

Line 135: Does the “partial data” not shown refer to the bands for the other genes? Could you include them in the supplement?

Figure 1: I am concerned about the dark lower half of this image. In the original, unedited version, the bottom half appears completely obscured. In the edited version it looks a bit better, but I would highly recommend rerunning the gel and getting a clear picture of the whole gel from the beginning (to avoid any questions about image manipulation). I would also highly recommend including a negative PCR control on the gel.

Line 138-139: Can you clarify whether S3CF is simply a positive control, and why you chose to use this strain? Please mention the inclusion of the positive control in the methods. Did you expect the strain you found to be closely related to this (and how did you identify the strain in your sample)?

Line 141-142: Are these sequences the ones you submitted to GenBank (not reference sequences you acquired from GenBank)? Please list these accessions in the data availability statement as well. Again, please address whether the S2CF strain has been previously described.

Line 156: Here < 670 bp is used as the common upper limit for NSP5 length. There have been four different lengths used at this point (666, 667, 651, 670). Is there any citation for a specific number being common, like 666, 667, or 651? Which of these is correct? Also NSP5 “genomes” should be “genes”.

Line 160: Can you specify here the repeat is a duplication? (I figured it out from the figure, but it wasn’t clear just reading the sentence).

Line 161: Is it correct that the other strains listed here have the duplication too? Can you state that a bit more clearly? Where are these strains from? What are the host species?

Figures 3 – 13 (all phylogenies): I would recommend mid-point rooting or including an outgroup sequence to aid visualization here. The images also look a bit compressed and low resolution. It would be useful to also include the actual support values (pertaining to clades with S2CF and close relatives) when discussing these trees in your results. I think it could also be helpful to think about other ways to visualize all of these trees, either combining them into a smaller number of figures or possibly including some in the supplementary, instead of including each one as a separate figure in the main text.

Figure 3: All the other trees have labeled clades. Are the clade labels missing from this figure?

Line 197: I’m not sure what this last sentence means. Is this addressing labeling and color coding clades? Could you clarify this?

Figure 5: I was confused about why some of the clades toward the bottom of the tree were so subdivided (sometimes with one strain per color), but is it correct that you choose how to color clades based on the genotype they have? Can you clearly state what the clade labels refer to in each tree.

Line 204-205: The “I” genotypes refer to VP6, right? Since this sentence is talking about VP1, it should say the “R1” clade, not “I1”, right?

Line 208-209: Can you clarify this close evolutionary relationship is specifically for VP1? In the other trees they often do not appear closely related. What evidence would suggest that strain YT is “human-like”?

Line 213-214: The “R” genotype is for VP1, not VP2, right? In your VP2 tree, the clades are labeled with “C”. “C” appears correct based on the information in the introduction.

Line 238-240: Are there other S2CF strains besides the one you found? Which ones are they?

Discussion

Line 298-300: This transition to aquaculture farms feels really abrupt, since you were sampling pigs and aquaculture was not previously mentioned. What does this have to do with the rotavirus you found? (if you could briefly recap the major finding before explaining the relevance of aquaculture, that would be helpful.)

Line 301-302: I think the possibility of human-pig cross species transmission should be mentioned in the introduction too (to help explain why this study is important). Are there any examples you can cite?

Line 313-315: Why does the S2CF genotype indicate it was transmitted from humans? It was closely related to porcine viruses in some trees too.

Line 353-354: Again, your sequence often clusters closely with other pig sequences in the phylogenies (for example, NSP1 and NSP2). I don’t think you can say definitively that the presence of this virus in pigs was due to human transmission. (Collecting human samples for analysis would definitely help clarify this, as you mention.) I do think it makes sense that the frequent clustering of human and pig RVA genes suggests there is cross-species transmission occurring though. (I also noticed in the abstract, you state that four genes clustered with humans, and state porcine-to-human transmission rather than human-to-porcine.)

Line 368-377: Can you connect these examples with your study? What is the significance of this information for interpretation of your results?

Comments on the Quality of English Language

There are many minor grammatical errors throughout the paper, but there are also a few sentences which need to be reworded to be understandable. For example, in line 331, I don't really understand what this is meant to convey: "...NSP5 gene fragment should be concerned mostly for its significant variability."

Author Response

Dear reviewer,

We appreciate all of your careful comments and professional suggestions.

We have revised the manuscript in accordance with the comments.

We also tried our best to meet the standards of required editorial
 corrections and have made all changes easily identifiable.

We hope that our revised manuscript meets your requirements.

Please see the attachment for the corresponding response in the point-by-point response.

Reviewer 2 Report

Comments and Suggestions for Authors

Rotaviruses (RVAs), belonging to the Reoviridae family, are the leading viral pathogens causing severe gastroenteritis in the young of both humans and many animal species worldwide. G4 is considered one of the most frequent G genotypes worldwide in humans, and is normally combined with the P[8] genotype and a Wa-like genogroup configuration. In the present study, the authors sequenced and analyzed the full-length of 11 segments of RVA/Pig-tc/CHN/S2CF/2023/G4P[6] strain detected in fecal samples collected in Guangdong Province, China in 2023. This study a unique 344-nt repeat sequence was first found in the non-encoding gene of NSP5. The results of the manuscript indicated the occurrence of potential interspecies transmission of porcine rotavirus. The article is well written, and the methodology used is adequate. The article is interesting and worthy of publication. The authors need to edit tables 3 and 4, because there is no red highlighting, and some of the text is not visible.

Author Response

(The authors gave the same response as above.)

Reviewer 3 Report

Comments and Suggestions for Authors

Dear Authors.

Thank you for the curated document. 

Figures 2 to 9 are distorted. Please fix before publication. 

Line 298 – “aquaculture” ??? Why is this relevant here?

Best regards, 

Author Response

(The authors gave the same response as above.)

Reviewer 4 Report

Comments and Suggestions for Authors

Dear Authors,

In this study, Zhou et al detected a distinct G4P[6] rotavirus strain, S2CF (RVA/Pig-tc/CHN/S2CF/2023/G4P[6]), in a piglet with severe diarrhoea in Guangdong, China. Genomic analysis showed a unique Wa-like genotype pattern and typical porcine RVA structure. Four gene segments (VP4, VP6, VP2, NSP5) were similar to human-like RVAs, suggesting potential porcine-to-human transmission. A novel 344-nt repeat sequence was found in NSP5.

I found your manuscript to be quite interesting and potentially valuable to the veterinary and One Health field. However, I believe it would greatly benefit from a thorough and critical review. I have made some suggestions for you below. I understand this may be a lot however, if you take time to go through these comments and make necessary corrections, it will be greatly appreciated.

Thank you.

Major concern

Point 1: Abstract

The abstract should be clear on the method used, specify which gene target RT-PCR was used to detect RV in pigs and mention briefly if the G-and-P genotype specific primers and protocol used prior to Sanger sequencing.

Also, note that it is not appropriate to use "genome" when referring to an individual gene. A genome represents the complete set of genetic material in an organism, while a gene refers to a specific segment of DNA. Please ensure to distinguish between "gene" and "genome" throughout the manuscript to enhance clarity and scientific accuracy.

Point 2 Literature review

The introduction provides a comprehensive overview of rotavirus classification and its genetic characteristics. However, it would benefit from the following suggestions:

1.    Include a summary sentence on human cases of rotavirus infection in China (beginning paragraph 1 where the authors reported cases in low-and-middle income countries).

2.    A clearer statement of the study's specific objectives and justification. While it outlines the genetic diversity and classification of RV, it lacks a clear explanation of the significance of the new findings. To improve, explicitly state why identifying and characterizing this special PoRVA strain is important, how it contributes to the existing body of knowledge, and its potential implications for disease control and prevention.

3.    Ensuring grammatical consistency and scientific accuracy throughout the text will enhance readability and credibility. For example, improving subject-verb agreement and clarifying the genetic classification of RVA strains will strengthen your introduction. Ensure that all sentences are grammatically correct and scientifically accurate. Eliminate redundancies and strive for conciseness to enhance readability and comprehension. I have provided suggestions under other comments, I hope this may be of interest to you!

Point 2: Methodology

For clarity and reproducibility, please address the following issues:

1.    Ensure the method includes details on the host from which the samples were collected.

2.    Clarify how samples were collected, whether they were collected rectally or from pen droppings. Ensure consistency in terminology (e.g., "faecal swabs" vs. "faecal samples").

3.    Could you kindly clarify and be clear in your presentation if all primers were designed for this study as per the method? Additionally, we suggest including PCR master mix details under your methodology or supplementary file, including primer concentration, template volume, and thermal cycling conditions, for enhanced reproducibility.

4.    For improved clarity, it is recommended that the sequencing methodology is detailed more explicitly. Specifically, please state that the complete gene sequences of RVA (n = 11) were achieved through sequencing amplicons from all rotavirus segments.

5.    Your method section is clear but needs to detail the sequence selection and preprocessing steps, including how sequences (including the number of sequences used) were selected, downloaded and whether they were trimmed to equal lengths post alignment. Ensuring these details are included will enhance the reproducibility and clarity of your phylogenetic analyses.

6.    Justify the use of both ML and NJ methods for phylogenetic analysis or consider selecting one method for simplicity.

7.    Section 2.4 Molecular dating analysis: The method described pertains to sequence alignment and mutation analysis rather than molecular dating analysis. For molecular dating, it is essential to include details on evolutionary models, calibration points, and software tools like BEAST or MrBayes used for estimating divergence times. This will align the method with standard practices in molecular dating analysis. I would suggest merging this content with section 2.3.

Point 3: Result presentation

For clarity and reader understanding, consolidate results into a coherent format separate from figures, ensuring a sequential flow, enhancing comprehension and accessibility for readers.

Sanger sequencing often produces sequences nearly twice the expected length due to errors like polymerase slippage and stem-loop structures. These can be mitigated by using designed primers, trimming sequences, direct sequencing without PCR purification, and re-sequencing with high coverage. Employing the shotgun assembly method and quality assessment is crucial. Your results are interesting; please clarify the methods used to address these sequencing errors.:

1.    Could you clarify how you edited and assembled the Sanger sequences? Specifically, did you reverse the reverse ABI file and assemble it to obtain a consensus sequence? Which software was used for this process, as MEGA might not support such tasks? Did you extract FASTA sequences directly from ABI files without assembly, and how did you determine the quality threshold?

2.    For example, following mapping of your Sanger reads to the assembled contigs and analysis of the raw BAM file, if the repeat sequence is due to an error, reads likely will not map to the repeated sequence. Kindly provide explanations for your findings, addressing how you handled potential sequencing errors and the specific methodologies employed to ensure accuracy. This will enhance the clarity and robustness of your results.

3.    General Comments on Phylogenetic Tree Presentation:

a.    Provide a plain tree with your reported sequence clearly highlighted. Once you provide your threshold value for bootstrap values, stating that circular node shapes represent bootstrap values ≥ 50 should suffice. This should apply to all trees in your study.

b.    I would suggest presenting one major phylogram in the main article, with the rest in a supplementary file for simplicity. I also suggest removing the colour codes for clades, as your study does not explain them clearly. If you retain colour coding, include a key similar to the one your created for bootstrap values. The interpretation of colour codes for accession numbers is unclear—are they by country, host, or other criteria? Kindly clarify on this.

c.     A critical observation of the phylograms suggests that some trees presented may be transformed cladograms (e.g., Figure 5), while others are proportional or untransformed trees (e.g., Figure 3). If this observation is correct, there is an inconsistency that raises questions about the reasoning behind these choices. Kindly clarify why different types of trees were used and ensure consistency across your figures.

Discussion

1.    To improve the manuscript, a thorough proofreading to correct grammatical errors and clarify scientific concepts is suggested. For example, vague terms like "abnormal fragments or genome" should be specified.

2.    Can you include a detailed discussion on the implications of the NSP5 gene insertion, particularly concerning viral replication and immune evasion?

Other comments

Line 18 - 24: The simple abstract may likely benefit from your additional review, for example replacing words like pathogens with “disease-causing agent”, novel reassortment with may be “new genetic mixing”, genome characteristics? etc.

Line 25 - 26: It looks like there is a disagreement here, do you mean “Diarrhoea in children, infants, and young animals around the world is caused by rotavirus.” Kindly review this sentence.

Line 26: kindly clarify this sentence, do the author’s mean “the associated zoonotic risk necessitates serious consideration of the complete genetic information of rotavirus.” Kindly clarify this sentence.

Line 27: Your clarification is much appreciated. I assume you are referring to a segmented genome rather than a gene and the emergence of a new viral strain rather than the formation of an entirely new virus. Typically, a new viral strain emerges rather than an entirely new virus. If this is indeed the authors' suggestion, could you kindly provide clarification?

Line 28 – 29: kindly review this sentence, the word at all time and help can be replace with a single, like essential for its prevention and control. Kindly reconsider reviewing this sentence.

Line 29 – 30 would designating this rotavirus strain using scientific names (e.g., Sus domesticus) be more appropriate?

Line 31 – 32 would it more appropriate to say whole genome sequencing and analysis of S2CF strain?

Line 34 would it be more appropriate to say, “clustered consistently with human-like RVAs, suggesting independent porcine-to-human interspecies transmission.” Kindly clarify this.

Line 36 I would suggest rephrasing this sentence to read “a unique 344-nt repeat sequence was identified for the first time in the non-coding region of NSP5.” Kindly clarify this sentence to specify that this sequence was discovered for the first time.

Line 38 The term "human" could be considered slightly ambiguous as it is not explicitly discussed in the abstract, though the authors does mention potential interspecies transmission to humans. To maintain clarity and relevance, you might consider replacing "human" with a term more directly associated with the findings, such as "interspecies transmission" or "zoonosis."

Line 41 I would use the word first and not firstly in this sentence. For example, “Rotavirus (RV), initially identified in the United States in 1969 [1], is a zoonotic pathogen associated with xxx.

Line 42 I would say “children under the age of five.

Line 44 Cause a large disease burden? I would reconsider reviewing this sentence, may be constitute or imposes a significant disease burden may be appropriate?

Line 44 – 45: I would reconsider revising the sentence “RV was classified into the genus 44 of Rotavirus within the family Reoviridae” to ensure clarity.

Line 48 Could you kindly clarify this sentence “consists of a triple-layered capsid that VP4 and VP7 were formed the outer capsid.” Is this sentence suggesting that RV consists of a triple-layered capsid, with VP4 and VP7 forming the outer capsid?" kindly address this issue?

Line I would suggest mentioning the number of RV groups currently identified. It might be appropriate saying RV is classified into n strains or groups including x, x, etc.

Line 55 – 56 I noticed that the hyperlink provided appears to be inaccessible or may direct users to a non-specific page. I would kindly suggest reviewing and rectifying this issue for the convenience of your readers.

Line 57 kindly correct this sentence; it appears the authors meant “Rotavirus group A (RVA) is considered the most pathogenic group for humans and pigs.” Kindly address this issue for clarity.

Line 61 kindly correct grammatical (plural form) and scientific error, this should rather read as “Gene rearrangement and reassortment of genomic segments have been continuously reported in RVA due to its segmented genome.” Kindly address this issue for clarity.

Line 62 – 64 I would avoid words like “In fact, all the” as used here, we can start a sentence with “. RVA strains are classified into three genogroups: Wa-like (genogroup 1), DS-1-like (genogroup 2), and AU-1-like (genogroup 3)”. Kindly address this issue for clarity.

Line 64 - 68 I would say Wa-like genogroup. In addition, this sentence is too long and would also benefit very additional review. For example, this piece of information is easier to be understood if rephrased as : “Most human RVA (HuRVA) and porcine RVA (PoRVA) strains belong to the Wa-like genogroup with a Gx-P[8]-I1-R1-C1-M1-A1-N1-T1-E1-H1 genotype constellation, while bovine RVA (BoRVA) strains belong to the DS-1-like genogroup with a G2-P[4]-I2-R2-C2-M2-A2-N2-T2-E2-H2 genotype constellation. Other animal RVA strains belong to the AU-1-like genogroup with a G3-P[9]-I3-R3-C3-M3-A3-N3-T3-E3-H3 constellation.” Kindly address this issue for clarity.

Line 69 - 72, I would suggest reviewing this sentence too. For example, remove the words “about the events and were,” and divide these sentences into two to read as below: “several studies on the reassortment or interspecies transmission of human and porcine RV strains are published annually. Examples include the PoRVA strain GDJM1 from China in 2022, the CN127 strain from China in 2020, the SO1199 strain from Japan in 2020, and the KCH148 strain from Kenya in 2019.” Kindly address this issue for clarity.

Line 73 I would avoid redundancy and informal expression like “of note” and simply present this sentence as "Reassortment or interspecies transmission events have not typically involved normally sized gene fragments." Kindly address this question.

Line 74, I can read this sentence as "Notably, the mutant NSP5 fragment, which is relatively conserved within specific species but often overlooked, has rarely been reported." Kindly address this issue for clarity.

Line 75 – 76, Do you mean “To date, the supershort NSP5 gene has only been documented in Japan [9], Belgium [10], and Indonesia [11]." Kindly address this issue.

Line 77 Do you mean "Most NSP5 gene fragments are approximately 666 bp in length, with fragments exceeding 666 bp being exceptionally rare, especially in China." Kindly address this question for clarity.

Line again, is this sentence mean “Only a rabbit rotavirus G3P[14], strain N5, has been described in China [12]." Kindly address this question for clarity.

Line 79 – 82 I would suggest avoiding redundancy and ensuring clarity. For example, these for example the summary from the four lines presented here simply means “genetic identification and characterization of an unconventional PoRVA strain was performed in this study.” Kindly address this issue to ensure clarity.

Line 85, I would remove the at the beginning of this sentence and start with “Faecal samples”. Kindly address this error.

Line 87, this should read “200 μL of the faecal swab.”  and “according to the manufacturer’s protocol.

Line 91 – 92 your sentence appears to mean: “Rotavirus A (RVA) detection was performed using a TaqMan probe-based real-time quantitative reverse transcription polymerase chain reaction (qRT-PCR) method developed in this study, targeting the NSP5 gene of RVA.” Please, kindly address this and avoid the use of the word “ourself” for clarity. In addition, an explanation for choosing NSP5 instead of the more commonly used NSP3 (for RVA detection) in the literature would be appreciated.

Line 96 Your sentence should be clarified as: “All primers and PCR thermal cycling conditions used in this study are detailed in Table 1.” Avoid using "were" and "procedure", be clear.

Line 99 Your sentence should be clarified as: “and Sanger sequenced by a commercial research laboratory ((BGI Tech Solutions (Beijing Liuhe) Co., Limited, Beijing, China)”. and not sequenced commercially.

Line 99 – 101, the researchers purified PCR products, cloned into the pMD19-T vector, and sequenced using M13 primers via the Sanger method. Could you please explain why this method was chosen over NGS (RNA) technology, especially given the virus's 18-19 kb size? Thank you.

Line 102 – 103, Please clarify what the length (bp) in Table 1 represents; typically, it should denote PCR product size. Additionally, the superscript "1" and "2" appear before "F" (forward) and "R" (reverse) in the table, but after these letters in the footnote.

Line 105, your sentence should be clarified as “constructed” and not conducted.

Line 108 – 109, your sentence should be clarified as: "The best-fit substitution models were selected based on the lowest Bayesian Information Criterion (BIC) scores as determined by the built-in model test application."

Line 110 – 111, your sentence should be clarified as: “To detect recombination, 11 aligned sequences of S2CF were analysed using RDP4 (version 4.101) with default settings across seven algorithms: RDP, GENECONV, Chimaera, MaxChi, BootScan, 3Seq, and SiScan."

Line 113 – 114, your sentence should be clarified as: "Recombination events were confirmed if at least four algorithms yielded significant p-values (P<1.0E-6)."

Line 115 – 121, I would suggest this section should be removed as the content is unlikely to reflect the title (kindly refer to my major comments under methodology).

Line 126, your sentence should be clarified as: “The eleven nucleotide sequences have been deposited in the GenBank database (https://www.ncbi.nlm.nih.gov/) and can be accessed under accession numbers xxx-xx.”

Line 131 – 132, I would suggest reviewing section 3.1. Your sentence should be clarified as: “For the positive sample, nearly full-length sequences of 11 segments were successfully obtained via RT-PCR and visualized by agarose gel electrophoresis.

Line 133 – 134, I would suggest “unexpectedly” not surprising. Your sentence should be clarified as: “than the expected size (651 bp) was observed for S2CF (RVA/Pig-tc/CHN/S2CF/2023/G4P[6]) (Figure 1).”

Line 130 – 139 (figure 10), Kindly specify the base pair size of the nucleic acid marker and clearly indicate your control samples in the methods section, including the source of the positive control. Review the gel image for potential visibility issues caused by dark edges. Consider presenting the image in the supplementary file and using a symbol to denote the S3CF positive control for clarity. Additionally, please review and refine the figure legend for precision.

Line 141, Your sentence should be clarified as: "We successfully obtained nearly full-length."

Line 142, your sentence should be clarified as "...and further characterized their similarity to reference strains."

Line 143 – 144: Your sentence should be clarified as: “As shown in Table 2, the VP7 gene is 1062 bp in length and shares the highest nucleotide sequence identity (96.07%) with the Chinese strain HLJ/15/1 from Heilongjiang.”

Line 145 – 147: Your sentence should be clarified as: “The VP4 (2359 bp) and VP6 (1356 bp) genes exhibit maximum nucleotide sequence identities of 96.40% and 95.87%, respectively, with the Chinese porcine-like human strain E931 (G4P[6]).”   

Line 148 – 150, your sentence should be clarified as: “The VP1, VP2, and VP3 genes were 3302 bp, 2717 bp, and 2527 bp in length, respectively, showing maximum nucleotide sequence identities (95.00%, 92.01%, and 96.87%) with the Chinese porcine strains YT(G4P[7]) and HeNNY-01(G4P[23]), and a human strain R479 (G4P[6]).”

Line 151 – 153 your sentence should be clarified as: “nucleotide sequence identities (96.03%, 95.50%, 95.63%, and 95.47%) with those of porcine and human strains.

Line 156 – 158 NSP5 gene, “sequence identity” and not identities here.

Line 159 your sentence should be clarified as: “was obtained through Sanger sequencing.”

Line 160 – 170 your sentence should be clarified as: “revealed an additional 344 bases from the 619th to the 962nd nucleotide at the 3’ UTR end, as also observed in Wa, Gottfried, DU2014-259, and CMH-N016-10.”

Line 162 – 164, kindly review this sentence, the rephrasing is unclear.

Line 166 – 168, your sentence should be clarified as: “The raw Sanger sequencing data and sequence alignments are provided in Supplementary Materials S2 and S3, respectively.” Please provide explanation if this is not the case.

Line 169 Going by the information provide in table 2, VP4 gene sequence obtained from this study was identical to Homo sapiens DR092 mitochondrion (16566 bp, accession number KT726034), not the VP4 gene sequence of 2359. Please correct this accession or clarify.

Line 170 173, the figure legend is likely to benefit f rom additional review and refinement.

Line 171 to 173, you should clarify these sentences, here is what I can understand: “The genetic characterization of strain S2CF encompassed analysis of each of its 11 gene segments using phylogenetic trees constructed from nearly full-length gene sequences. Reference sequences were sourced from the GenBank database.”

Line 178 – 179, kindly review this sentence.

Line 189 – 193, kindly review the figure legend, exclude information on nucleotide best fit models from figure legends as this is clearly provided in your method. The figure legend for phylogenetic trees should be summarized. You should clarify your legend as: “Maximum likelihood phylogenetic tree based on VP7 coding sequences, calculated with 1,000 bootstrap replicates. Node labels represents bootstrap values ≥ 0.5 are shown. The S2CF strain isolated in this study is marked with a blue triangular branch tip shape.

Line 203 – 207, you should clarify this sentence as: “For the inner structural proteins, analysis of the VP1 gene revealed sequence identities ranging from 85.4% to 95%. The phylogenetic analysis of VP1 indicated that strain S2CF formed a distinct cluster alongside several Chinese porcine isolates (the I1 clade), such as JS, YT, and CN127, distinct from bovine, human, feline, and pigeon Rotaviruses (RVs) (Figure 6). Interestingly, strains JS (G5P[23]) and CN127 (G12P[7]) represent reassortant Rotaviruses between porcine and human strains, with a Wa-like backbone” Please, kindly review to enhance scientific accuracy.

Line 208 – 217, kindly review your result presentation as suggested for line 203 – 207.

Line 251 – 257, kindly consider a thorough review of the result presented under this section ensuring grammatical and scientific accuracy. For example, table 3 cross referenced in the text is provided in the supplementary file and should be clearly mentioned as a supplementary table.

Line 298 – 300, you should clarify your sentence: do you mean, Rotavirus infections pose significant challenges to aquaculture farms worldwide, particularly in developing countries that have faced substantial losses due to the virus.

References

Kindly update manuscript citations with current literature.

Comments on the Quality of English Language

Kindly refer to the attached peer-review report

Author Response

(The authors gave the same response as above.)

Round 2

Reviewer 1 Report

Comments and Suggestions for Authors

This version of the manuscript is much improved. See a few more comments to address below:

Line 24-25: Is it really accurate to say diarrhea is "mainly" caused by rotavirus? I think it would be better to say rotavirus is "a major cause" of diarrhea.

Line 62: This is helpful, but "P" is missing from the following genotypes: G3[8], G4[8], G9[8], G12[8]

Line 94-97: This is a helpful addition, but I think since S2CF is the name of one of your samples (and not a previously mentioned strain from the introduction) it doesn't make sense to reference the specific strain's name this way here. It might be better to say something like "We aimed to characterize novel rotavirus strains within porcine fecal samples and investigate the potential for cross-species transmission between pigs and humans."

Line 100: Can you clarify the samples were denoted S2CF and S3CF so it is more clear why you named the strains this way?

Line 166: Can you please also define the S3CF strain here, clarifying that the NSP5 gene was the expected length?

Line 170 (gel): S3CF doesn't appear to be shown here. Did you upload a new original gel? I'm not seeing it here (the one on the review website doesn't match this new figure). It would be ideal if the gel in the paper did show S3CF alongside S2CF and the positive and negative controls.

Line 287-288: Can we really assume "every" child under 5 has encountered the virus? I went to Reference 26, but in their paper it isn't clear where they got this information either. Could you cite a couple studies that actually give more precise estimates based on their own (or clearly cited) data instead?

Line 293: I think here, "unexpectedly" is meant to be "respectively."

Line 320-321: I still don't think you can say this "showed it was a human-to-porcine interspecies transmission event." In the trees that you say show clustering with humans (VP6 and NSP5, for example), it also appears to be closely related to other pig-derived viruses. I think if you depict your trees as cladograms instead (not using branch lengths) these relationships will be much easier to see. Also, keep in mind there are likely other pig or human rotaviruses that haven't been sequenced but might actually be closer relatives of S2CF. I think it would also be useful to include the S3CF genes in your trees to see if they appear similar to any of the S2CF ones.

Line 359-360: Regarding "we monitored the strain of S2CF was a porcine-origin strain containing human genes with a rearranged NSP5 gene," I think you should say containing genes with similar to human and porcine rotaviruses, rather than "human genes."

Line 383-385: I'm confused by what this is saying: "In this study, only the VP4 and VP7 genes were amplified, we could not be able to discover the diversity of other genes, especially in NSP5 gene." You were able to amplify and sequence all 11 genes, right? Does "this study" refer to a different study?

Comments on the Quality of English Language

Most English errors are not serious, but there are many throughout the paper, sometimes affecting the reader's ability to understand what the authors are really trying to say. Editors will need to assist in editing for clarity and grammatical correctness before publication.

Author Response

Dear Reviewer 1, 
We greatly appreciate all the comments and suggestions you have provided for our manuscript. We have give our second corresponding response in the point-by-point response letter and carefully revised it according to your comments (Round 2), and uploaded the new original gel (Figure 1). The amendments are highlighted in red for round 2 in the re-revised manuscript.

We hope that our re-revised manuscript meets your requirements. 

Best Wishes.
